# Design and Fabrication of a Thin and Micro-Optical Sensor for Rapid Prototyping

**DOI:** 10.3390/s23177658

**Published:** 2023-09-04

**Authors:** Nobutomo Morita, Wataru Iwasaki

**Affiliations:** Sensing System Research Center, National Institute of Advanced Industrial Science and Technology (AIST), Tosu 841-0052, Japan

**Keywords:** optical sensing, heterogeneous integration, optical integration, blood monitoring

## Abstract

Optical sensing offers several advantages owing to its non-invasiveness and high sensitivity. The miniaturization of optical sensors will mitigate spatial and weight constraints, expanding their applications and extending the principal advantages of optical sensing to different fields, such as healthcare, Internet of Things, artificial intelligence, and other aspects of society. In this study, we present the development of a miniature optical sensor for monitoring thrombi in extracorporeal membrane oxygenation (ECMO). The sensor, based on a complementary metal-oxide semiconductor integrated circuit (CMOS-IC), also serves as a photodiode, amplifier, and light-emitting diode (LED)-mounting substrate. It is sized 3.8 × 4.8 × 0.75 mm^3^ and provides reflectance spectroscopy at three wavelengths. Based on semiconductor and microelectromechanical system (MEMS) processes, the design of the sensor achieves ultra-compact millimeter size, customizability, prototyping, and scalability for mass production, facilitating the development of miniature optical sensors for a variety of applications.

## 1. Introduction

Optical sensing has several advantages owing to its non-invasiveness and high sensitivity [1]. However, optical sensors tend to be relatively large because they are composed of various optical components, such as a light source, lenses, optical fibers, filters, and detectors. The miniaturization of optical sensors will free them from spatial and weight constraints and enable them to be portable, wearable, embeddable, and implantable [2,3,4,5,6], thereby expanding their range of applications [7,8,9,10]. The scope of the term miniaturization ranges from desktop size to hand size to grain size. Reflective optical sensors, which consist of a light-emitting source, such as a light-emitting diode (LED), and a light-receiving part, such as a photodiode (PD), have a relatively simple component configuration and are particularly advantageous for grain-size miniaturization. An example of a fully integrated reflective optical sensor is the sensor module for the saturation of percutaneous oxygen (SpO_2_), ADPD144RI (Analog Devices, Inc.) [11], which is highly integrated with amplifiers, drivers, and communication interfaces in addition to optical elements. However, application-specific integrated circuits (ASICs) generally have high development costs, and it is difficult to change their specifications, such as making them thinner or changing the photosensitive area. Another example of a reflective optical sensor module, NJR5510R (Nisshinbo Microdevices Co., Ltd.) [12], has only a light emitter and receiver, and it can be developed at a lower cost than ASICs or complementary metal-oxide semiconductor (CMOS)-based optical sensors. However, the total volume of the system, including the preamplifier, resistors, capacitor, printed circuit board (PCB), and wiring, limits miniaturization because these external electrical components must be prepared near the optical components. Under these restrictions, a compact and thin yet highly prototypical and customizable design is required to extend the benefits of optical measurements over a wider area.

This paper describes the design and fabrication of an optical sensor that combines highly prototypical and customizable properties with small and thin total sizes. The monitoring target of the sensor was set to thrombi in an extracorporeal circuit membrane oxygenation (ECMO) as a case study. In extracorporeal circulatory devices such as ECMO, thrombi can occur at various locations including pumps, artificial lungs, and connector gaps, causing serious complications such as thromboinfarction [13,14]. Early thrombus detection is an old challenge, and several methods have been used to detect thrombi in specific parts of ECMO circuits, such as spectral cameras for blood pumps, optical coherence tomography for connector gaps, and ultrasound analysis for blood pumps [13,14,15]. Such monitoring requires the monitoring of many points, such as tubing with a diameter of approximately 10 mm, blood pumps, artificial lungs with free-form surfaces, narrow internal flow paths, and narrow gaps between these components. Therefore, it is desirable for the sensors that detect thrombi to be as thin and as small as possible. We set the target size of the developed optical sensor to within a 5 mm × 5 mm area and 1 mm thickness, including the base circuit board, which we aimed to install at sites facing a major thrombus risk.

## 2. Materials and Methods

### 2.1. Basic Design of the Optical Sensor

Figure 1a shows a schematic of the proposed sensor structure. The sensor arrangement and light path are depicted as examples of thrombus detection in a connector. The light emitted from the light-emitting diodes (LEDs) passes through the blood and thrombus and is received by the photodetector (PD) in the sensor. When a thrombus occurs, the hematocrit (Hct) inside the thrombus changes [16], and the thrombus is detected based on the change in absorbance in the light path. A shading block and CMOS chip with a PD and signal amplification circuit were fabricated to realize a small and thin prototype sensor. Additionally, LEDs, a multilayered ceramic capacitor (MLCC), and a shading block were surface-mounted on the CMOS chip (Figure 1b). The CMOS chip and LEDs were electrically connected to a flexible board via wire bonding. All components, including the bonding wires, were molded simultaneously using epoxy resin. The design, based on component mounting on the CMOS chip and batch molding, contributed to the miniaturization of the overall sensor package.

### 2.2. CMOS Chip

The design of the CMOS chip is illustrated in Figure 2. The CMOS chip comprises monolithically fabricated photodiodes, amplifier circuits, and mounting pads for LEDs and a capacitor chip. The outline size of the CMOS chip is 2900 μm × 2900 μm, and the chip is 400 μm thick. The active area of the PDs is 600 μm × 600 μm. One PD was covered with a top metal layer, and the differential output from the other was used as the PD output. The output of the PD was amplified using amplifier circuits with two different gains, H-gain and L-gain, and the output pin was selected based on the light-receiving intensity. The CMOS chip contains three 355 μm × 355 μm LED pads. By mounting the LEDs with different emission wavelengths, optical measurements for up to three different wavelengths could be performed by switching the LEDs to be driven. The wiring pads for all external connections, such as the PD outputs, supply voltage, ground, and LEDs, were arrayed on the top edge of the CMOS chip. As LED chips have anode and cathode terminals (shown as A+ and K− in Figure 2b) on the top and bottom surfaces, respectively, the anode side is connected via an aluminum wiring pattern on the CMOS from the LED mount pad, while the cathode side is connected by wire bonding from the LED top electrode to the cathode pad.

Generally, when integrated circuits (ICs) such as amplifiers are used, bypass capacitors are used together to ensure operational stability against sudden voltage fluctuations and noise [17]. This is because fabricating capacitors with sufficient capacitance on ICs is difficult. Therefore, we propose a structure in which the MLCC is mounted on a CMOS chip to ensure good performance and miniaturization. Assuming that an MLCC with size code 0201 (0.6 × 0.3 × 0.2 mm) is to be mounted, 200 × 300 μm MLCC mounting pads were designed on the CMOS chip.

The CMOS chip was fabricated based on the process rules of minimum line and space of 6 μm/4 μm and minimum via diameter of 2 μm. By adopting a process rule with such a large margin and using a part of the chip as the mounting surface for the electrical components, the prototyping cost can be reduced to less than a few tenths of the cost of a 0.35 nm CMOS process while providing sufficient functionality.

### 2.3. Shading Block

In the structure of the developed optical sensor, it is crucial to consider that the LED emission is not directly received by the PD because of the close distance between the LEDs and the PD, and their location on the same surface. Therefore, we designed a 250 μm thick shading block with a pore diameter of 600 μm for light transmission, which selectively receives light through the blood. The shading block was fabricated by processing Si wafers via a deep reactive ion etching (DRIE) process called the Bosch process [18,19,20]. The DRIE process selectively etched Si vertically while alternately supplying C_4_F_8_ (passivation gas) and SF_6_ (etching gas). The metal mask for DRIE was formed via lithography on a Si wafer with sputtered aluminum, followed by dry-etching patterning of aluminum. The process conditions were optimized according to Ref. [21]. All fabrication processes were implemented using a minimal fab, which is a semiconductor fabrication system that excels in prototyping and high-mix, low-volume production [22]. Figure 3 displays a photograph of the shaded blocks. Eight shading blocks were fabricated on a half-inch Si wafer. Shading blocks composed of Si with the DRIE process enabled us to easily achieve a high processing accuracy of micrometer precision and the same smoothness as that of a CMOS chip, which enabled stacking without gaps on the CMOS chip.

### 2.4. Assembly

An outline of the sensor assembly is illustrated in Figure 4. The CMOS chip was mounted on a flexible substrate using wire bonding pads. Three LEDs, a 0.1 μF MLCC, and the shading block were bonded on the CMOS chip by using Ag conductive paste (Panasonic, DBC-183SG). The sensor was baked at 150 °C for 10 min on a hotplate to cure the Ag paste. Three LEDs with different emission wavelengths, 660 nm (Epister, ES-SABRPN14D), 810 nm (KLV, EOLC-810-25), and 940 nm (Epistar, ES-SAUFPN14), were chosen for the fabrication, and they differed in their absorbance characteristics with respect to blood and dependence on oxygen saturation [23]. Electrical connections from the CMOS chip to the flexible substrate-bonding pads and topside electrode pads of the LEDs were achieved using wire bonding. The LEDs and wiring were encapsulated using a potting clear epoxy resin (Inabata & Co., Ltd., Osaka, Japan, IK-0010) and curing at 150 °C for 4 h.

### 2.5. Evaluation of Light Detection Performance

The range of the detectable light intensity of the optical sensor was evaluated to determine its applicability. The evaluation was performed by recording the output voltage when light of a known intensity was incident on the PD. Figure 5 illustrates the evaluation system. Two of the developed sensors were used for evaluation: one as the source of the test light and the other as the receiver. These sensors were fixed face-to-face using a jig at distance Z, such that the light-receiving and light-emitting surfaces faced each other. Z was set to 14 mm for the 660 nm wavelength measurement and 6 mm for the 810 nm and 940 nm measurements, based on prior verification of the maximum distance at which the sensor output *V_out_* could saturate. The current applied to the LED was varied from 0 mA to 10 mA in 0.1 mA increments. *V_out_* was recorded for 1 s at 1 kS/s, and the average *V_ave_* and noise root mean square (RMS) *V_rms_* were calculated for each LED current. The light intensity versus LED current was measured using a power meter (Thorlabs, PM400, and S130VC), and the LED current was calibrated to the light intensity at each wavelength and Z. The LED control, voltage recording, and calculations were carried out using an I/O module comprising a laptop and a LabVIEW DAQ system composed of a module mounting chassis (National Instruments, cDAQ-9179), current control module (National Instruments, NI-9203), and analog input and output module (National Instruments, NI-9205). The measurable light intensity range was defined as the range where *V_rms_* was smaller than 0.01 V and *V_ave_* was not saturated.

### 2.6. Optical Measurement Test of Blood

Porcine blood was used as the sample because its optical properties in terms of scattering and absorption are similar to those of human blood [24]. Porcine blood was obtained during slaughter and was provided by Tokyo Shibaura Organ Co., Ltd. (Tokyo, Japan). Blood samples with adjusted Hct and oxygen saturation (SaO_2_) were prepared to monitor the blood condition using the sensor. Hct was adjusted to 100%, 75%, and 50% of its initial concentration by diluting with saline (Otsuka Pharmaceutical Co., Ltd. (Tokyo, Japan), Otsuka normal saline). The SaO_2_ was controlled by supplying pure oxygen gas from a gas cylinder via a membrane oxygenator (Nagayanagi Co., Ltd., (Tokyo, Japan), Nagasep M30-B). Blood samples were placed in a clear-colored case (Merck KGaA, Greiner CELL STAR T-25 flask) made of polystyrene (PS), which has a high and flat transmission rate from 500 to 1600 nm of the wavelength range [25]. The Hct and SaO_2_ of the blood samples were measured via centrifugation [26] and a blood gas analyzer (Siemens Healthcare K.K., (Tokyo, Japan), Epoc blood analysis system), respectively. An optical measurement setup was constructed to evaluate the performance of the developed sensor (Figure 6). The developed sensor was placed at the bottom of a polystyrene case with a wall thickness of 1.0 mm. Each LED in the developed sensor was turned on sequentially to measure the PD’s light-receiving intensity at each wavelength. In addition, to evaluate the shading effect of the shading block, the PD output was measured without the polystyrene case and blood, and all the LEDs were turned off. The illumination current of the LEDs was adjusted to 3 mA for the 660 nm LED and 10 mA for the 810 nm LED and 940 nm LED. The blood samples were fully stirred via inverted agitation prior to measurement, and *V_out_* was recorded after 30 s of placement on the sensor to ensure that the blood movement was sufficiently stable. Each blood sample was analyzed three times. *V_out_* was normalized to the light intensity *L_λ_* for each wavelength using Equation (1).
(1)$$L_{\lambda }=\frac{V_{out}-V_{0}}{V_{std,\lambda }-V_{0}},$$
where *V*_0_ is the output voltage at incident light intensity 0, *V_std_*_,*λ*_ is the output voltage during the measurement of the sample before Hct and SaO_2_ adjustment, and *V_out_* is the output voltage during each blood sample measurement.

## 3. Results and Discussion

The optical sensor was fabricated using the aforementioned process. The total size of the sensor is 3.8 mm × 4.8 mm in area and 0.75 mm in thickness, including the thickness of the flexible wiring substrate and epoxy mold. The results of the light detection performance are shown in Figure 7. The output voltages at incident light levels of 0 and *V*_0_ were 0.73 V. At all wavelengths, *V_out_* increased in proportion to the light intensity, decreasing linearly above 4.6 V and saturating at 5 V. *V_rms_* was large (over 0.1 V) at zero light power, but it decreased as the light intensity increased and stabilized at 5 mV or lower when *V_out_* was 1.2 V or higher. Therefore, the upper and lower limits of measurable intensity at each wavelength were determined from 1.2 V to 4.6 V as the measurable range (Table 1).

The initial Hct of the blood samples used for the blood assay test was 42%. Dilution with saline resulted in deoxygenated blood samples with Hct values of 42%, 31%, and 22%, and oxygenated blood samples with Hct values of 42%. The corresponding oxygen saturation levels were 69.1%, 66.8%, 64.3%, and 96.2%, respectively. The illumination current of the LEDs was set at 3 mA for the 660 nm LED and 10 mA for the 810 nm LED and 940 nm LED. The results of the optical measurements are shown in Figure 8. The PD outputs were normalized to each wavelength using the PD output at 42% Hct in deoxygenated blood using Equation (1). Each value and error bar shows the average and standard deviation of the three measurements. The intensity of each wavelength when varying from 22% to 42% of Hct decreased with increasing Hct. Sakota et al. reported a large decrease in Hct level (from 12 to 4 g/dL in hemoglobin concentration) due to thrombus formation [16]. Because hemoglobin is present in red blood cells, the hemoglobin concentration in the blood is proportional to Hct, indicating that the Hct of a thrombus can be as low as 1/3 of the Hct of the blood. Thus, this change in the optical intensity can be applied to thrombus detection. Comparing the intensity at each wavelength in the oxygenated and deoxygenated blood of 42% Hct, the intensity at 660 nm increased 3.0 times, whereas slight changes occurred at 810 nm and 940 nm (1.03 and 0.86 times, respectively). This result is consistent with the trends of the absorbance of oxygenated and deoxygenated blood depicted in [23], indicating changes in the optical properties of the blood. The change in the PD output when the LED was turned on and off without placing any object on the sensor was less than 0.2% of the blood sample measurement output at all wavelengths. Therefore, there was no stray light inside the sensor, and the results indicated that the shading block provided sufficient light shielding.

The proposed sensor design has the potential to customize its characteristics and achieve high performance. By changing the LEDs to LEDs with different emission wavelengths, the measurement wavelength can be easily changed without additional components such as color filters, spectrometer units, or additional manufacturing processes. The advantage of this concept is that the design of the CMOS chip, which is most affected by specification changes, does not need to be changed. By choosing LEDs with appropriate wavelengths for new measurement targets or applications and assembling them with pre-prepared components such as CMOS chips and flexible substrates, rapid prototyping of customized specifications is possible. LED manufacturers offer a wide selection of bare chips in sizes that can be mounted on the CMOS chip’s LED mounting pad (355 μm square), with wavelengths ranging from ultraviolet (UV) to shortwave infrared (SWIR). For example, EPIGAP OSA Photonics GmbH has over 120 bare LED chips in the chip size range of 235–365 square micrometers, with 59 different wavelengths from 355 nm to 1720 nm as catalog products [27]. The photosensitivity can be controlled by changing the hole diameter of the shading block. These advantages aid in the experimental optimization of the performance in the development of optical sensors. For CMOS chips, the thickness can be reduced to a few tens of micrometers by polishing the back side. Although the cost will increase, additional functions, such as LED drivers and communications, can be integrated into the chip without changing the size by using a submicrometer-order CMOS process. Additionally, the side surface of the sensor can replace the dicing surface because it has no function except as a wire bonding area. If the electrical contacts are extracted from the bottom surface using silicon instead of wire bonding, the sensor design can be adapted for wafer-level packaging. These additional options for CMOS chips can enhance performance without modifying the optical design obtained during development and have the potential to contribute to advancing the seamless development from proof of concept to commercialization.

## 4. Conclusions

A small, thin, and prototypical sensor was developed, and the customizability of its design and its potential for miniaturization and multifunctionality were discussed. The structure of the sensor can be customized by changing the measurement wavelength and photosensitivity through changing its components, and has various scalabilities, such as further thinning, miniaturization, and scaling for mass production via wafer-level packaging.

## Figures and Tables

**Figure 1 sensors-23-07658-f001:**
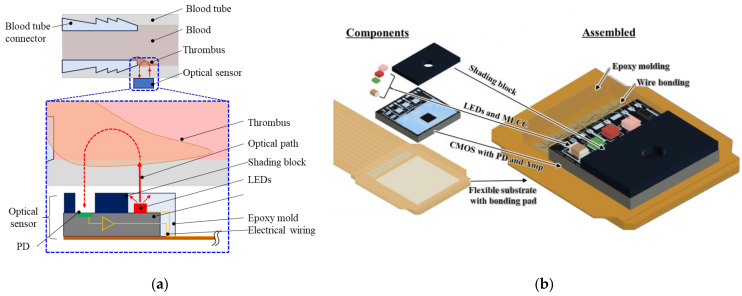
Schematic of the micro-optical sensor design: (**a**) cross-sectional view of the optical sensor and example installation on a connector gap for thrombus sensing; (**b**) proposed sensor components and their integration.

**Figure 2 sensors-23-07658-f002:**
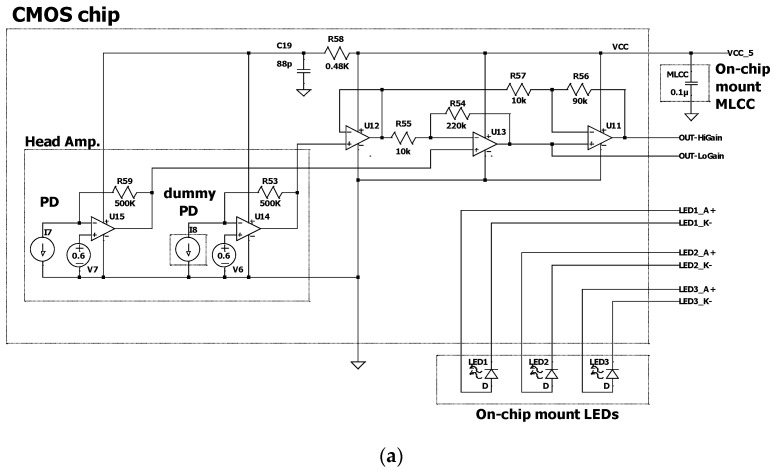
CMOS-processed sensor base: (**a**) circuit diagram of the PDs and amplifier circuit; (**b**) layout overview of the CMOS chip; (**c**) photograph of the CMOS chip. The dummy PD is covered using the top metal layer.

**Figure 3 sensors-23-07658-f003:**
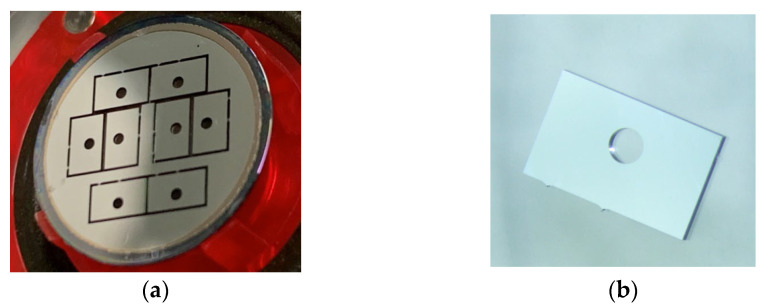
Pictures of Si-etched shading blocks fabricated using the minimal fab: (**a**) Shading blocks processed on a half-inch Si wafer. (**b**) Chip of the shading block with a 600 μm diameter and 250 μm depth hole.

**Figure 4 sensors-23-07658-f004:**
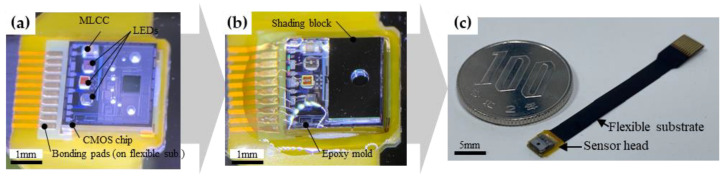
Outline of the sensor assembly: (**a**) CMOS chip, LEDs, and MLCC integration. (**b**) Epoxy-molded sensor. (**c**) Completed optical sensor.

**Figure 5 sensors-23-07658-f005:**
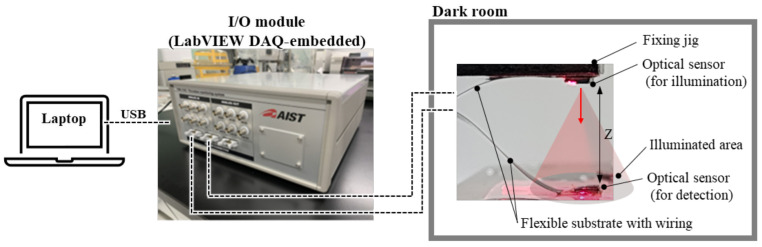
Setup for evaluation of light detection performance of the sensor.

**Figure 6 sensors-23-07658-f006:**
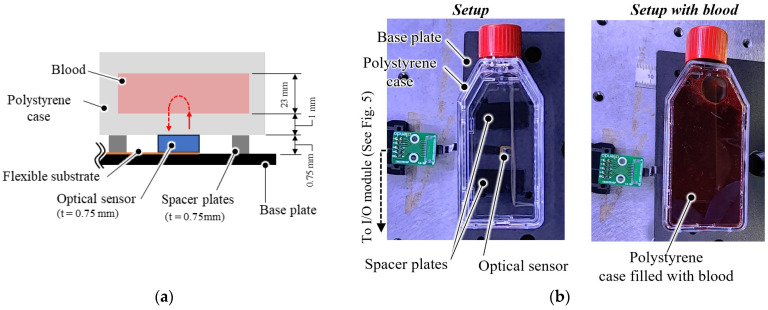
Experimental setup of optical measurement test: (**a**) Schematic of the setup. (**b**) Pictures of the setup without and with blood sample.

**Figure 7 sensors-23-07658-f007:**
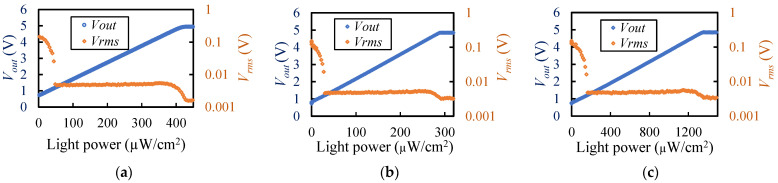
Results of the light detection performance of the developed sensor: (**a**) 660 nm. (**b**) 810 nm. (**c**) 940 nm.

**Figure 8 sensors-23-07658-f008:**
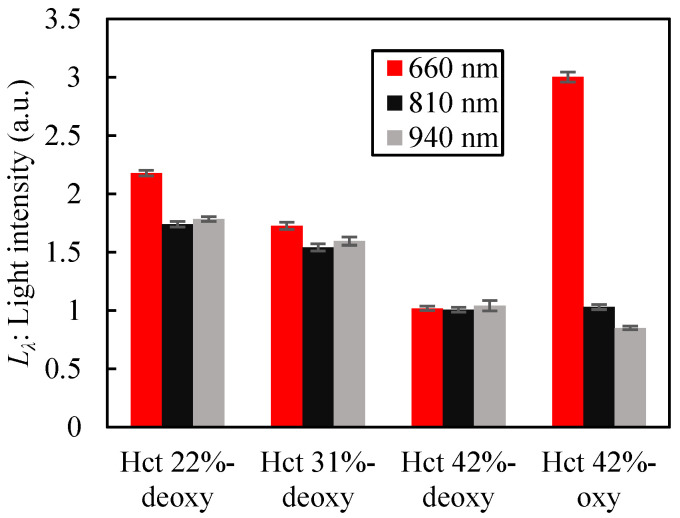
Results of optical measurement test of blood.

**Table 1 sensors-23-07658-t001:** Measurable range of light intensity and light intensity corresponding to *V_rms_*.

Wavelength (nm)	Min. Detection Limit (µW/cm^2^)	Max. Detection Limit (µW/cm^2^)	*V_rms_* (V)	*V_rms_* in Optical Power (µW/cm^2^)
660	30	271	0.00493	0.350
810	48.3	386.1	0.00496	0.487
940	164.6	1271	0.00496	1.605

## Data Availability

All data were disclosed in the figures and a table.

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
