# Peer review of "Design and Fabrication of a Thin and Micro-Optical Sensor for Rapid Prototyping"

_sensors, 2023, doi:10.3390/s23177658_

Round 1

Reviewer 1 Report

Authors introduce the optical sensing world using a trivial approach. I suggest to improve this part with more scientific items. They present a project od thin optical sensor "all included" with LED, Amplifier and Photo Detector. Very simple project. Anything new? The CMOS circuit is not particularly complex. The authors show a picture of their chip and draw a prototype setup. A real setup with pictures, power, blood and so on could clarify if this project is still an idea or a true circuit. I suggest to add details and a real case. the change of the LED to investigate other wave lengths is a good idea but it doesn't seem so simple. You should provide more information.

some good ideas are inside this paper but you should improve the description of the case study to have a paper for Senors.

References should also be improved.

Reviewer 2 Report

This manuscript is rough and cannot be accepted in its current form.

1) The methodology in this study lacks sufficient detail to allow for evaluation of the validity of the results. It is important to provide a step-by-step description of the experimental procedures, including the design and fabrication of the 3 optical sensors, and the monitoring protocol.

2) The results and discussion section are too short and only has one figure that displays all the results of optical measurement, which is seriously inadequate for the evaluation of the monitoring performance of CMOS-IC sensor. On the other hand, figure 6 is not standard, the legend should be inside the box and the error bar also missed.

The language needs to be edited extensively.

Reviewer 3 Report

The work presented here is novel, and the writing style of this paper is engaging and concise, making it easy to follow the author’s line of reasoning. The authors have analyzed their results and pointed out the direction for future work.

Some additional comments:

In this manuscript, a thin and miniatured optical sensor based on a CMOS chip was developed. The authors successfully demonstrated its functionality by detecting the PD outputs at 3 different Hct in deoxygenated blood for three wavelengths. Their work realized the miniaturization of optical sensor at low-cost and expanded the applications of the optical sensor to the medical field. Additionally, they took advantage of the semiconductor and MEMS processes, and made it easy for mass production. Optical sensor’s miniaturization and low-cost manufacturing attract lots of attention. The methods used in this work will help to motivate the development of other optical sensors for various applications.   

The work presented here is novel, and the writing style of this paper is engaging and concise. The authors have analyzed their results and the conclusions drawn from their measurement are supported by the evidence. The references are appropriate.

I would suggest making a minor revision for publication. Specific comments are given below:

1.     For the first time in the article, it will be good if the authors can put the acronym in parentheses after the full term. And later you can stick to using the acronym.

For example, CMOS in line 27, SpO2 in line 29, ASICs in line 36, and PD in line 60, et al.  

2.     Could the authors confirm if there is a typo in line 42? “an extracorporeal circuit membrane oxygenator”

3.     Could the authors provide more explanation for the 0.1 uF MLCC? What is its usage? Is it the C18 in the Figure 2 (a)?

4.     It will be good if the authors can make it clear that the three K- pads on the CMOS chip are for the bonding the topside electrode of the LEDs.

Round 2

Reviewer 1 Report

A good paper with interesting items to use optical sensors. The possibility to change the sensor remain a weak point. Maybe a future version can improve this aspect. 

Reviewer 2 Report

The revised manuscript could be accepted now.